# Avoidance of Natural Suckling from Dams with Bovine Leukemia Virus Is a Low Priority Countermeasure against Postnatal Transmission

**DOI:** 10.3390/vetsci8110255

**Published:** 2021-10-27

**Authors:** Hirohisa Mekata, Eriko Kusuda, Chiho Mori

**Affiliations:** 1Center for Animal Disease Control, University of Miyazaki, 1-1 Gakuen-Kibanadai-Nishi, Miyazaki 8892192, Japan; 2Hokusatsu Veterinary Clinical Center, Kagoshima Prefecture Agricultural Mutual Aid Association, 13-1 Todorokicho, Satsuma, Kagoshima 8951813, Japan; e_kusuda@nosai-hokusatu.jp; 3Department of Clinical Examination, Mori Animal Hospital, 90-5 Hosono, Kobayashi 8860004, Japan; chihohoi0624@hotmail.com

**Keywords:** artificial rearing, BLV, colostrum, enzootic bovine leukosis, natural suckling, postnatal transmission

## Abstract

Although natural suckling from dams with bovine leukemia virus (BLV) has not been recommended in Japan, the frequency of BLV transmission through natural suckling under natural conditions is still unclear. The purpose of this study was to elucidate the risk of BLV transmission through natural suckling. Dams with BLV were classified into three groups (high, middle, low) based on the proviral loads (PVLs). PCR positivity of their colostrum and the correlations between the ratios of calves with BLV and types of feeding milk were analyzed. In dams with low PVLs, no colostrum or calves were confirmed to have BLV. In dams with middle and high PVLs, 17 out of 25 (68.0%) colostrum were PCR positive, and 10 out of 23 (43.4%) and 13 out of 29 (44.8%) calves with natural suckling and artificial rearing were infected with BLV, respectively. No difference was confirmed between the infection rates of natural-suckled and artificially reared calves. Thus, we concluded that the avoidance of natural suckling from dams with BLV and the introduction of artificial rearing were low priority countermeasures against BLV transmission.

## 1. Introduction

Bovine leukemia virus (BLV), a member of the *Retroviridae* family and the *Deltaretrovirus* genus, is an etiological agent of fatal B-cell leukemia and malignant lymphoma in cattle, which are, together, known as enzootic bovine leukosis (EBL). Although more than 95% of BLV-infected cattle remain EBL-free for life, one to five percent of such cattle develop EBL several years after infection [1,2]. Australia, New Zealand and many Western European countries have successfully eliminated BLV. However, BLV continues to spread across the world, and high seroprevalence has been confirmed in many countries including Japan and the USA [3,4]. BLV establishes lifelong infections, and no vaccines or therapeutic agents are available for preventing BLV infection or EBL development. Thus, preventing infection in cattle is the only measure for reducing EBL.

The dam-to-calf transmission of BLV includes prenatal and postnatal transmission. Prenatal transmission can occur in utero, and postnatal transmission can occur through natural suckling [5,6]. We previously reported that prenatal transmission was detected in 18.4% of newborn calves born from dams with BLV [7]. In this previous study, all newborn calves were immediately separated from their dams after delivery and fed with pasteurized colostrum and milk replacer to prevent transmission through natural suckling. The frequency of postnatal transmission under natural conditions is still unclear. Colostrum from dams with BLV contains both BLV-infected lymphocytes and neutralizing antibodies against BLV, and newborn calves ingest both [8,9]. Studies have warned that there is a risk of BLV transmission through natural suckling such as for human T-cell leukemia virus type 1 (HTLV-1) [10,11]. HTLV-1 is the causative agent of a fatal T-cell leukemia in humans, and it is closely related to BLV [12]. HTLV-1 infection is more prevalent among breastfed children than bottle-fed children [13,14]. Japan has performed nationwide screening of pregnant women and promoted bottle feeding for children born from mothers with HTLV-1 [15], and this policy has helped to reduce the number of HTLV-1 infections. Thus, artificially rearing Japanese livestock such as by feeding pasteurized colostrum, colostrum replacer and milk replacer, which are treated by heating, drying and freezing, is recommended. On the contrary, some studies have suggested that natural suckling reduces the frequency of BLV transmission under natural conditions [16,17].

In our routine tests, the BLV infection rates were not markedly different between naturally suckled and artificially reared calves. Therefore, we strongly suspect that, in nature, BLV transmission through natural suckling is infrequent. The immunity of newborn calves is completely dependent upon maternal colostrum [18]. A deficiency of maternal colostrum in newborn calves increases diarrhea and respiratory disease [19]. Although colostrum replacer contains immunoglobulins, natural suckling is still desirable because endemic pathogens vary among farms, regions and countries. In addition, artificial rearing increases the financial burdens and workloads of farmers. Thus, we need to clarify the risk of BLV transmission through natural suckling in nature.

## 2. Materials and Methods

### 2.1. Animals and Samples

In this study, the feeding of pasteurized colostrum, colostrum replacer and milk replacer are collectively referred to as artificial rearing in order to distinguish them from natural suckling. This study took place on four beef cattle production farms (Japanese Black cattle) in Japan in the Miyazaki and Kagoshima prefectures. Sample group 1 was collected on Farms A and B to clarify the correlations among BLV infections in newborn calves, the PVLs of dams and the PVLs in colostrum. A total of 40 pairs of dams and calves from Farm A (*n* = 28) and Farm B (*n* = 12) were sampled. All of the sampled dams had confirmed BLV infections. Blood and colostrum samples were collected from the dams within 12 h after delivery. In addition, blood samples were collected from the newborn calves within 12 h after delivery and at 1 month old, to diagnose potential BLV infection and measure PVLs. The calves fed with udder milk were weaned by 10 days old. After weaning, all the calves were fed with milk replacer. Sample group 2 was collected from Farms C and D in order to clarify the correlations among BLV infections in calves; the PVLs of dams; and the types of feeding milk including udder milk, pasteurized colostrum and colostrum replacer (First Start 60, JAPAN Nutrition, Tokyo, Japan; Calfsupport Dash, Zenoaq, Koriyama, Japan). The pasteurized colostrum used in this study was collected from dairy farms with BLV in the same areas as the study farms and was checked to contain anti-BLV antibody by enzyme-linked immunosorbent assay (ELISA). The colostrum was frozen and pasteurized at 60 °C for 30 min before use [20,21]. In addition, we analyzed the neutralizing antibody titers against BLV in calves. A total of 81 pairs of dams and calves from Farm C (*n* = 50) and Farm D (*n* = 31) were sampled. Forty-nine out of 81 dams had confirmed BLV infections. A total of 16, 13, 28 and 6 calves were fed with udder milk from dams with BLV, pasteurized colostrum, commercial colostrum replacer and udder milk from dams without BLV, respectively. In addition, 18 calves were fed with both udder milk and commercial colostrum replacer in order for them to ingest enough immunoglobulin. Blood was collected from the dams within 24 h of delivery and from calves at 1, 4 and 7 months old. The calves fed with udder milk were weaned by 3 months old. After weaning, all the calves were fed with milk replacer. All the calves had confirmed colostrum intake within 24 h of delivery.

### 2.2. ELISA and Real-Time PCR

All the blood samples were tested for anti-BLV antibodies and BLV proviral DNA using ELISAs and real-time PCR, respectively. The ELISAs were performed using a BLV gp51 antibody detection kit (Nippon Gene, Tokyo, Japan) according to the manufacturer’s instructions. The real-time PCR was performed using a LightCycler 96 System (Roche Diagnostics, Indianapolis, IN, USA). Genomic DNA was extracted from 0.2 mL of whole blood using a fully automated system (magLEAD system, Precision System Science, Chiba, Japan). The DNA concentration was determined using a NanoDrop 8000 spectrophotometer (Thermo Fisher Scientific, Waltham, MA, USA), the DNA samples from the blood were diluted to 20 ng/μL and the samples with consistent concentrations of genomic DNA were used for real-time PCR. A probe qPCR mix (TaKaRa Bio, Kusatsu, Japan) and PrimeTime qPCR assays (Integrated DNA Technologies, Coralville, IA, USA) including target-gene-specific primers and probes were used to quantify the BLV PVLs and detect the bovine β-actin gene. The primer and probe sets used in this study have been described in detail in previous reports [22,23]. Each amplification procedure was performed in duplicate, and the results are expressed as the numbers of proviral copies per 50 ng of genomic DNA. BLV positivity in dams and calves was determined using the results of ELISAs and real-time PCR, respectively.

### 2.3. PVL Classification of Cattle with BLV

We previously reported that cattle with fewer than 100 BLV copies/50 ng were hardly able to horizontally transmit the virus to BLV-free cattle [24]. However, dams with more than 2000 BLV copies/50 ng could easily vertically transmit the virus to their calves [7]. Based on these reports, the PVLs of dams were assumed to correlate with the risk of BLV postnatal transmission. Therefore, we classified the BLV-infected dams into three groups as follows: “High PVL” (more than 2000 BLV proviral copies/50 ng); “Middle PVL” (100–2000 copies/50 ng); and “Low PVL” (less than 100 copies/50 ng).

### 2.4. Detection and Quantification of Proviral DNA in Colostrum

Genomic DNA was extracted from a mixture of 0.2 mL of suspended colostrum to 50 μL of elution buffer using the magLEAD system. The parts of the colostrum with high consistency were diluted at most 2-fold with normal saline before the DNA extraction. One microliter of extracted DNA was used for the real-time PCR mentioned above without DNA dilution, and the concentrations (copies/mL) were determined. The PVL levels in the colostrum (copies/mL) were calculated as follows: the concentration of the extracted DNA (copies/mL) × 0.05 (mL)/0.2 (mL) × the dilution ratio for the colostrum. The efficiency with which proviral DNA was extracted from colostrum was assumed to be 100%.

### 2.5. Neutralizing Antibody Titers in Calves

The titers of neutralizing antibodies against BLV were quantified using the syncytium-induction inhibition assay described in a previous report [8]. The neutralizing antibody titers are expressed as the reciprocals of the highest serum dilutions that showed a positive reaction.

### 2.6. Statistical Analysis

A chi-squared test was used to compare the PCR positivity of the colostrum and the PVL group of dams. Fisher’s exact test was used to compare the PCR positivity of calves and types of milk feeding. The D’Agostino–Pearson normality test was used to verify the normal distribution of the PVLs in the calves at 1, 4 and 7 months old. All the PVLs were distributed normally. Therefore, Tukey’s multiple comparisons test was used to compare the mean PVLs with one another. These analyses were performed using the GraphPad Prism 6 software (GraphPad Software, San Diego, CA, USA). *p* < 0.05 was considered statistically significant in this study.

## 3. Results

To evaluate the risk of BLV transmission through colostrum, we measured the pro-virus in colostrum from dams with BLV. Twelve out of 13 (92.3%) and five out of 12 (41.6%) samples of colostrum from dams with high and middle PVLs were PCR positive, respectively (Table 1). On the other hand, no colostrum (0/15) from dams with low PVLs was PCR positive. The median (average) PVLs in the colostrum from dams with high and middle PVLs were 2250 (2999) and 1500 (1500) copies/mL, respectively. Significant differences among the PVL groups were observed, as was a higher rate of PCR positivity in the colostrum from dams in the higher PVL group (*p* < 0.05).

To evaluate the risk of postnatal transmission through natural suckling, calves born from dams with BLV were tested for BLV infection at 1 month old. No calves born from dams without BLV (*n* = 32) or with low PVLs (*n* = 35) were infected with BLV, regardless of whether they were naturally suckled or artificially reared (Table 2). In the case of natural suckling, 60.0% (6/10) and 30.7% (4/13) of the calves born from dams with high and middle PVLs were infected with BLV, respectively. On the other hand, in the case of artificial rearing, 46.6% (7/15) and 50.0% (8/16) of the calves born from dams with high and middle PVLs were infected with BLV, respectively. No significant differences in the BLV infection rates between calves naturally suckled and artificially reared were observed (*p* = 0.68, high PVL; *p* = 0.45, middle PVL, respectively). We confirmed that all of the 1-month-old calves without BLV were both ELISA- and PCR-negative at 7 months old with one exception, which was thought to have been infected with BLV via a horizontal route.

We confirmed that the BLV infection rates between naturally suckled and artificially reared calves were not significantly different. On the other hand, studies have reported that natural suckling from dams with BLV reduces the number of calves with BLV in nature [16,17]. This might suggest that the maternal antibody against BLV has the potential to protect calves from BLV infection. Thus, we measured the neutralizing antibody titers using the syncytium-induction inhibition assay in BLV-uninfected calves fed with udder milk, pasteurized colostrum and colostrum replacer at 1, 4 and 7 months old. The neutralizing antibody titers were markedly different among calves fed with udder milk from dams with BLV, calves fed with pasteurized colostrum and those fed with colostrum replacer (Figure 1a). About half of the calves fed with udder milk from dams with BLV or pasteurized colostrum had neutralizing antibody titers over four at 1 month old. On the other hand, only one calf fed with colostrum replacer had a neutralizing antibody titer over four. Most of the neutralizing antibody titers in 4-month-old calves were less than two (Figure 1b).

The BLV PVLs in cattle were stable after the acute phase, and cattle with higher PVLs represented a higher risk of horizontal transmission [24,25]. We confirmed that the median (average) PVL in 0-day-old calves was 51.0 (58.5) copies/50 ng in sample group 1. On the other hand, the median (average) PVL in the dams of these calves was 2011 (2155) copies/50 ng. This suggests that PVLs in calves increase as they grow. Therefore, we tried to reveal the fluctuation of PVLs in 14 calves vertically infected with BLV at 1, 4 and 7 months old. The PVLs of 1-month-old calves ranged from 86 to 1746 copies/50 ng, and the median value was 560 copies/50 ng (Figure 2). The PVLs tended to increase, ranging from 83 to 4747 copies/50 ng (median: 1129 copies/50 ng) at 4 months old and were stable at 7 months old (range: 199 to 5259 copies/50 ng; median: 1269 copies/50 ng). There were significant differences between PVLs at 1 and 4 months old and between 1 and 7 months old (*p* < 0.05).

## 4. Discussion

This report discusses BLV transmission through postnatal routes under natural conditions. We revealed that dams with low PVLs hardly transmitted BLV to their newborn calves. In addition, approximately 40% of newborn calves born from dams with high and middle PVLs were infected with BLV regardless of artificial rearing. We concluded from these results that the avoidance of natural suckling and the introduction of artificial rearing to all newborn calves born from dams with BLV was not an efficient countermeasure against BLV transmission.

In this study, 92.3%, 41.6% and 0% of the colostrum samples from dams with high, middle and low PVLs were PCR positive, respectively (Table 1). More than 95% of the breastmilk from mothers of HTLV-positive children has been shown to be PCR positive [26]. Thus, natural suckling from dams with low PVLs presented a very low risk of BLV transmission. A previous study reported that 73.3% of breast milk from mothers with HTLV-1 was PCR positive, and the PVLs were, at most, 119,000 copies/mL [26]. This study reported that 42.5% of the colostrum from dams with BLV was PCR positive, and all of the PVLs were lower than 8500 copies/mL. The PVLs in colostrum from dams with BLV might be lower than the PVLs in milk from mothers with HTLV-1. No differences in BLV infection rates between calves that were naturally suckled and artificially reared from dams with middle and high PVLs were observed (Table 2). In addition, approximately 40% of calves born from dams with middle and high PVLs were infected with BLV regardless of artificial rearing. Although we could not conclude that natural suckling from dams with middle and high PVLs presented no risk of BLV transmission, we conclude that dams with low PVL hardly transmitted BLV through delivery and natural suckling. The BLV infection rate in calves was strongly affected by the PVLs of dams [7]. Some previous studies have not considered the PVLs of dams, leading to opposite results regarding the postnatal transmission of BLV [6,16].

BLV antibody titers of 16 and 64 were reported to completely protect against BLV infection in experimental infections [27,28]. In this study, about half of the 1-month-old calves fed with udder milk and pasteurized colostrum from dams with BLV had neutralizing antibody titers of more than four (Figure 1). This is not enough to completely prevent BLV infection but could partially prevent horizontal infection. PVL quantification is important because cattle with higher PVLs have a higher risk of transmitting BLV [24]. The median values of PVLs were changed from 560, 1129 and 1269 copies/50 ng in 1-, 4- and 7-month-old calves, respectively (Figure 2). Based on our results, the segregation of calves with BLV by the age of 4 months old is desirable.

Whether the natural suckling of newborn calves from dams with high and middle PVLs presents a risk of BLV transmission is still unclear. Previous studies have proven that lambs and calves (not newborn calves) were easily infected with BLV when fed with udder milk from dams with BLV [6,29]. Although the udder milk included both BLV-infected lymphocytes and anti-BLV antibodies, these animals could not incorporate the maternal antibodies into their bodies. The reaction between BLV-infected lymphocytes and maternal antibodies in newborn calves is still unclear.

HTLV-1 infection is more prevalent among breastfed children than bottle-fed children [13]. Although BLV is closely related to HTLV-1, we conclude that the avoidance of natural suckling and the introduction of artificial rearing are not effective countermeasures against BLV infection. Some studies have reported that short-term breastfeeding (for less than 3 or 6 months) results in significantly lower seroconversion rates for HTLV-1 infection [30]. Although the natural suckling periods in calves are different depending on the feeding style, country and species of cattle, most Japanese calves are weaned by 3 months old. In addition, the proviral DNA in milk from dams with BLV was found to be very low or undetectable after 7 days post-delivery [8]. As far as we know, long-term fluctuation of the PVLs in milk from mothers with HTLV-1 has been not reported. However, breast milk samples from an average of 11 days post-delivery (range, 5–47 days) were used for PVL quantification, and 73.3% of them were PCR positive [26]. Thus, the positive period for the provirus in milk might be longer for mothers with HTLV-1 than for dams with BLV. Although future studies are required, these differences might affect the differences in risk between BLV and HTLV-1 transmission through direct feeding.

## 5. Conclusions

We revealed that dams with low PVLs hardly transmitted BLV to their newborn calves through natural suckling under natural conditions. In addition, approximately 40% of newborn calves born from dams with high and middle PVLs were infected with BLV regardless of artificial rearing. Artificial rearing increases the financial burdens on and workloads of farmers, and natural suckling is desirable for the health of calves. Although we need to clarify the risk of BLV transmission through natural suckling from dams with high and middle PVLs, we conclude that the avoidance of natural suckling from dams with BLV and the introduction of artificial rearing to newborn calves are low priority countermeasures against BLV transmission.

## Figures and Tables

**Figure 1 vetsci-08-00255-f001:**
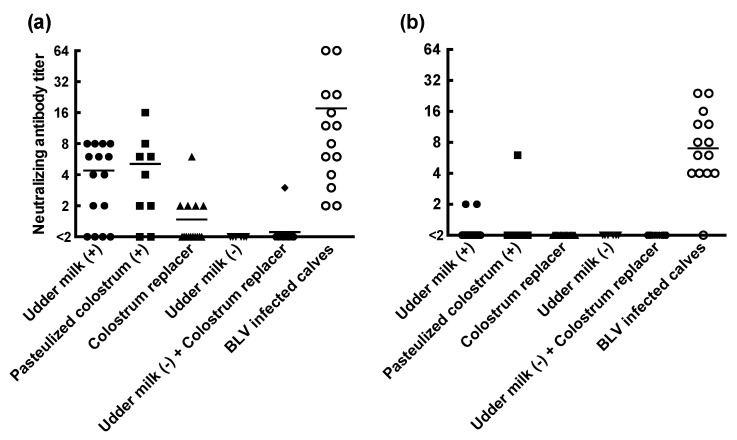
Titers of neutralizing antibodies against BLV in 1- and 4-month-old calves fed with udder milk, pasteurized colostrum and colostrum replacer. The titers of neutralizing antibodies against bovine leukemia virus (BLV) in 1 (**a**)- and 4 (**b**)-month-old calves were compared between the types of feeding milk. The titers were quantified using the syncytium-induction inhibition assay. The types of feeding milk and the number of sampled calves were as follows: Udder milk (+): udder milk from dams with BLV (*n* = 15); Pasteurized colostrum (+): pasteurized colostrum from a nearby farm with BLV (*n* = 9); Colostrum replacer: commercial colostrum replacer pasteurized using a drying treatment (*n* = 19); Udder milk (−): udder milk from dams without BLV (*n* = 6), Udder milk (−) + colostrum replacer: udder milk (−) supplemented with commercial colostrum replacer (*n* = 18); BLV-infected calves: 1-month-old calves vertically infected with BLV (*n* = 14). Horizontal lines show the median values of the neutralizing antibody titers.

**Figure 2 vetsci-08-00255-f002:**
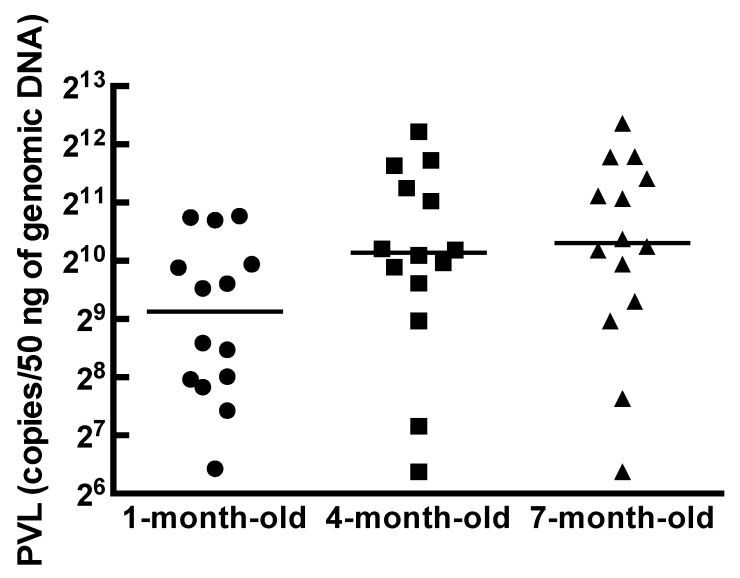
Changes in BLV PVLs in calves vertically infected with BLV. Bovine leukemia virus (BLV) proviral loads (PVLs) in 14 calves vertically infected with BLV were monitored at 1, 4 and 7 months old. The median PVLs at 1, 4 and 7 months old were 560, 1129 and 1269 copies/50 ng, respectively. Horizontal lines show the median values of PVLs. Significant differences were observed between PVLs at 1 and 4 months old and between 1 and 7 months old (*p* < 0.05).

**Table 1 vetsci-08-00255-t001:** Positivity and proviral loads (PVLs) of bovine leukemia virus (BLV) in colostrum.

	BLV Status in Sampling Dams		
	High PVL ^a^	Middle PVL ^a^	Low PVL ^a^
PCR positivity	92.3% (12/13)	41.6% (5/12)	0% (0/15)
Range of PVL (copies/mL of colostrum)	750–8500	500–2500	-
Median (average) value of PVL (copies/mL of colostrum)	2250 (2999)	1500 (1500)	-

^a^ High PVL: more than 2000 copies/50 ng; Middle PVL: 100–2000; Low PVL: fewer than 100.

**Table 2 vetsci-08-00255-t002:** BLV positivity in 1-month-old calves fed with udder milk, pasteurized colostrum and colostrum replacer.

Sample Group	Types of Feeding Milk	BLV Status in Dams of Sampled Calves			
		High PVL ^a^	Middle PVL ^a^	Low PVL ^a^	BLV (-)
1	Natural suckling	60.0% (6/10)	37.5% (3/8)	0% (0/10)	-
	Artificial rearing ^b^	0% (0/3)	50.0% (2/4)	0% (0/5)	-
2	Natural suckling	-	20.0% (1/5)	0% (0/11)	0% (0/24)
	Artificial rearing	58.3% (7/12)	50.0% (6/12)	0% (0/9)	0% (0/8)
Total	Natural suckling	60.0% (6/10)	30.7% (4/13)	0% (0/21)	0% (0/24)
	Artificial rearing	46.6% (7/15)	50.0% (8/16)	0% (0/14)	0% (0/8)

^a^ High PVL: more than 2000 copies/50 ng; Middle PVL: 100–2000; Low PVL: fewer than 100. ^b^ Artificial rearing: feeding with pasteurized colostrum or colostrum replacer.

## Data Availability

The data that support the findings of this study are available from the corresponding author, H.M., upon reasonable request.

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
