# Peer review of "Avoidance of Natural Suckling from Dams with Bovine Leukemia Virus Is a Low Priority Countermeasure against Postnatal Transmission"

_vetsci, 2021, doi:10.3390/vetsci8110255_

Round 1
Reviewer 1 Report
The overall idea of this research project is sound and could provide useful information to avoid excessive labor in feeding beef calves vs. allowing them to nurse in an attempt to minimize risk of BLV transfer.
I am not sure how to interpret the data or if the data was analyzed in a way to make the conclusions that are presented here. It does not seem normally distributed to compare and the statistical analysis needs to be checked by a statistical expert.
My specific comments are below:
Veterinary Sciences
Avoidance of Direct Breastfeeding from Dams with BLV is a low priority countermeasure against postnatal transmission
Title: Consider changing title to “Avoidance of direct nursing of Dams with Bovine Leukemia Virus is a low priority countermeasure against postnatal transmission.”
Overall throughout the paper, breastfeeding should be changed for “nursing”.
Abstract:
- The results sentences are not clear what PVL are considered high, medium, and low. It is difficult to understand what the low PVL transmission rate is and what “hardly” is referring to. This is further explained in the methods but the abstract verbiage needs to stand alone.
Materials and Methods:
2.1 Animals and Samples
- What is the definition of “soon?” after birth? Within 4 hours? 24 hours?
- Were the calves in the different groups tested for successful passive transfer of maternal antibodies?
2.2 ELISA and Real-Time PCR
- How were the assays validated? Was this done in a research laboratory or an accredited veterinary diagnostic laboratory?
- PVL is estimated from extracted DNA from an automated extraction system and quantified via nanodrop? Is this directly related to the number of viable viral particles?
- Was every batch of pasteurized milk tested for BLV? Was milk sterile or pasteurized? The processing temperature and time suggest pasteurization. Was the milk tested before and after pasteurization to ensure the equipment was working properly?
2.5 Neutralizing antibody titers in calves
- Were serum neutralization assays validated in the lab in any way or performed by an accredited veterinary diagnostic laboratory?
Results
- In the materials and methods, the authors state that all dams were BLV positive. In the results BLV negative dams are reported.
- Define how a calf is considered BLV positive. Based on ELISA, PCR, or both? If ELISA is part of the diagnosis for calves fed colostrum and nurse from a BLV positive dam, the assay will have interference from maternal antibodies. The statement of confirmation of BLV infection rates is a discussion, not a result.
- Is the data normally distributed? Group 1 as 10 nursing calves and 3 bottle fed calves and Group 2 only has 12 in the bottle fed group for the high PVL group.
- What is the relevance of the results in the text between table 2 and table 3, including table 3? The amount of IgG was not measured in the colostrum vs. pasteurized colostrum vs. colostrum replacer and it is well known that processing colostrum will destroy IgG. The data is nicely summarized and presented, but there is no interpretation as to what the author’s think it means.
- What statistical analysis is used for data in figure 2? Are the PVL’s statistically different at different ages?
Author Response
Dear Reviewer 1,
Thank you very much for making time in your busy schedule. We are grateful for your valuable comments to improve our manuscript. Our responses are written below. The revised words are marked in red font.
Reviewer 1’s comments and our responses
Comment#1. It does not seem normally distributed to compare and the statistical analysis needs to be checked by a statistical expert.
Respons#1. We reconsidered the statistical analysis of our data. Our data was not complicated and it was not difficult to analyze using a biostatistics software.
Commen#2. Title: Consider changing title to “Avoidance of direct nursing of Dams with Bovine Leukemia Virus is a low priority countermeasure against postnatal transmission.”
Response#2. Thank you for your comment. Another reviewer indicated to correct all the words “breast feeding”, “breast” and “bottle feeding” to “natural suckling", “udder” and “artificial rearing”, respectively. Thus, we revised the title to Avoidance of Natural Suckling from Dams with Bovine Leukemia Virus Is a Low Priority Countermeasure against Postnatal Transmission.
Comment#3. Overall throughout the paper, breastfeeding should be changed for “nursing”.
Response#3. Thank you for your comment. Please refer to Response #2.
Comment#4. The results sentences are not clear what PVL are considered high, medium, and low. It is difficult to understand what the low PVL transmission rate is and what “hardly” is referring to. This is further explained in the methods but the abstract verbiage needs to stand alone.
Response#4-1. To clear whose PVL were considered as high, medium, and low, we added some words as a follow.
Line 15: Dams with BLV were classified into three groups (high, middle, low) based on the proviral loads (PVLs), and PCR positivity of their colostrum and the correlations between the ratios of calves with BLV and types of feeding milk were analyzed.
Response#4-2. To clear the meaning of “low PVL transmission rate” and “hardly” in this manuscript, we revised the sentences as a follow.
Line 20-21: No difference was confirmed between the infection rates of natural suckled and artificial reared calves.
Comment#5. What is the definition of “soon?” after birth? Within 4 hours? 24 hours?
Response#5. Following the reviewer’s comment, we revised some words as a follow.
Line 76-77: …from the dams within 12 hrs after delivery. In addition, blood samples were collected from the newborn calves within 12 hrs after delivery…
Line 93: …the dams within 24 hrs after delivery and …
Comment#6. Were the calves in the different groups tested for successful passive transfer of maternal antibodies?
Response#6. Following the reviewer comment, we added a sentence as a follow.
Line95-96: All the calves were confirmed colostrum intake within 24 hrs after delivery.
Comment#7. How were the assays validated? Was this done in a research laboratory or an accredited veterinary diagnostic laboratory?
Response#7. This assay was done in a research laboratory. This ELISA kit is commercially available and used for many public institutions in Japan.
Comment#8. PVL is estimated from extracted DNA from an automated extraction system and quantified via nanodrop? Is this directly related to the number of viable viral particles?
Response#8. Following the reviewer comment, we added some words as a follow.
Line 106-107: … and the samples with consistent concentration of genomic DNA were used for real-time PCR.
Comment#9. Was every batch of pasteurized milk tested for BLV?
Response#9. Following the reviewer comment, we added some words as a follow.
Line 84-85: as the study farms and was checked to contain anti-BLV antibody by ELISA.
Comment#10. Was milk sterile or pasteurized? The processing temperature and time suggest pasteurization.
Response#10. Following the reviewer comment, we revised all the words “sterilized” to “pasteurized”.
Comment#11. Was the milk tested before and after pasteurization to ensure the equipment was working properly?
Response#11. Yes.
Comment#12. Were serum neutralization assays validated in the lab in any way or performed by an accredited veterinary diagnostic laboratory?
Response#12. This assay was done in a research laboratory.
Comment#13. In the materials and methods, the authors state that all dams were BLV positive. In the results BLV negative dams are reported.
Response#13. In sample group 1, all the dams were BLV-positive. However, in sample group 2, we used both BLV positive and negative dams. To avoid confusion, we added a sentence as a follow.
Line 88-89: Forty-nine out of 81 dams had confirmed BLV infections.
Comment#13. Define how a calf is considered BLV positive. Based on ELISA, PCR, or both? If ELISA is part of the diagnosis for calves fed colostrum and nurse from a BLV positive dam, the assay will have interference from maternal antibodies.
Response#13. We used both ELISA and PCR for BLV diagnostic test, and we finally diagnosed BLV infection in a calf by result of real-time PCR. As the reviewer pointed out, ELISA possibly detected maternal antibodies.
To clear the meaning of this sentence, we added a word as a follow.
Line 113-114: BLV positivity in dams and calves was determined using the results of ELISAs and real-time PCR, respectively.
Comment#14. The statement of confirmation of BLV infection rates is a discussion, not a result.
Response#14. I’m sorry I can’t understand the meaning of this suggestion.
Comment#15. Is the data normally distributed? Group 1 as 10 nursing calves and 3 bottle fed calves and Group 2 only has 12 in the bottle fed group for the high PVL group.
Response#15. We used a Fisher’s exact tests for this statistical analysis. Because the number was not the random variable, we need not to care about normal distribution.
Comment#16. What is the relevance of the results in the text between table 2 and table 3, including table 3? The amount of IgG was not measured in the colostrum vs. pasteurized colostrum vs. colostrum replacer and it is well known that processing colostrum will destroy IgG. The data is nicely summarized and presented, but there is no interpretation as to what the author’s think it means.
Response#16. Thank you for your comment. As the reviewer pointed out, table 3 was not related to the hypothesis which was the maternal antibody might have the potential to protect calves from BLV infection. Thus, we deleted table 3 and some sentences in result and discussion sections.
Comment#17. What statistical analysis is used for data in figure 2? Are the PVL’s statistically different at different ages?
Response#17. Following the reviewer’s comment, we performed a statistical analysis and added some sentences in “Materials and Methods”, “Result” and “Table legend” sections as follows
Line 140-143: The D’Agostino-Pearson normality test was used to verify the normal distribution of the PVLs in the calves at 1, 4 and 7 months old. All the PVLs were distributed normal. Therefore, Tukey's multiple comparisons test was used to compare the mean PVLs with one another.
Line 208-210: There were significant differences between PVLs at 1 and 4 months old and between 1 and 7 months old (p < 0.05).
Line 215-217: Significant differences were observed between PVLs at 1 and 4 months old and between 1 and 7 months old (p < 0.05).

Reviewer 2 Report
Using the terms, "breast feeding","breast" and "bottle feeding" are inappropriate for bovines and are human terms.
Please change them all in the paper. My suggestions would be
"Naturally suckled" for "breast feeding"
and
"Artificially reared" for "bottle fed"
Change "breast" to "Udder"
Author Response
Dear Reviewer 2,
Thank you very much for making time in your busy schedule. We are grateful for your valuable comments to improve this manuscript. Our responses are written below.
Reviewer 2’s comments and our responses
Comment: Using the terms, "breast feeding", "breast" and "bottle feeding" are inappropriate for bovines and are human terms. Please change them all in the paper. My suggestions would be "Naturally suckled" for "breast feeding" and "Artificially reared" for "bottle fed" Change "breast" to "Udder".
Response: Thank you for your comment. We revised all the above words correctly in the revised manuscripts including figures and tables.
